# Influence of Water Saturation on Curtain Formation of Frozen Wall

Hewan Li, Jian Liu *, Laigui Wang and Tianjiao Ren

College of Mechanics Engineering, Liaoning Technical University, No. 47, Zhonghua Road, Xihe District, Fuxin 123000, China
* Correspondence: lj13142rtj@163.com

**Abstract:** In the process of in situ mining of underground oil shale, it is necessary to build an underground frozen wall around the mining area to prevent the inflow of groundwater and prevent oil and gas leakage. However, the water saturation of the frozen soil affects the formation of the freezing circle. In practical engineering, when building a frozen wall in soil under different water saturation conditions, the curtain is often not closed. It also affects the construction and operating costs of the entire project. In this paper, the methods of laboratory experiments and numerical simulation analysis are supplemented by the analysis of ordinary differential equations and partial differential equations. Aiming at the lowest cost, the temperature drop process and freezing time around the frozen well under different water saturation conditions are calculated and analyzed in detail. The formation law of the frozen wall curtain under different water saturation conditions is obtained. It provides a reference for the setting of the low-temperature loading time and the oil shale extraction range of frozen wells for different soils in practical engineering.

**Keywords:** underground frozen wall; water saturation; freezing circle; finite-element modeling; freezing time; numerical simulations

## 1. Introduction

Due to the increasing shortage of ground space resources, the development of under-ground space resources has attracted more and more attention. Among them, underground oil shale is a very important resource and is very rich in underground storage. After various technical processes, shale oil can be produced from oil shale as the main energy material. Commonly used processing means include both ground dry distillation and underground in situ mining [1,2]. The ground dry distillation method is mainly to extract oil shale from the underground space. After crushing and other processes, it is placed in the distillation equipment for heating, and finally, shale oil is formed [3]. Compared with other technologies, ground dry distillation is more mature. However, there is still a lot of latent heat in semi-coke and other substances after dry distillation, (However, after dry distillation, emi-coke and other substances still produce a large amount of latent heat,) which is often less effectively utilized and discarded [4]. The purpose of in situ mining technology is to directly heat oil shale stored underground and make oil shale directly into shale oil or shale gas, then extract it through the well [5,6]. This technique is suitable for oil shale layers buried in deep and complex environments and can effectively avoid some defects in ground dry distillation. Therefore, in situ mining technology has gradually become the main way to mine oil shale.

In the process of in situ high-temperature underground mining, in order to effectively prevent the leakage of oil shale and shale gas and isolate groundwater around the mining area, artificial underground is necessary to freeze in the mining area. The United States Shell company used this technique for the first time in 2006 for underground in situ mining to form a 4900 m² underground frozen wall curtain and efficiently exploit the oil shale [7],

eliminate the effects of surrounding groundwater, and also prevent waste from occurring during in situ mining to spread outside the mining area and pollute the environment [8]. When forming a freeze wall curtain, multiple freezing pipes are used to create a low-temperature environment to freeze the surrounding strata and form multiple cylinder freezing areas, namely frozen wall closures [9,10].

Current research shows that different factors have different effects on the cost, time required, or isolation effect of forming the frozen wall curtain. (1) In terms of freezing temperature, Guo and Huo (1999) [11] constructed a mechanical model of the temperature field and studied the development of the peak displacement of the temperature field. Zhai et al. (2015) [12], through finite-element simulation, found that the temperature of the frozen wall under different water saturation conditions is most appropriate from −10 °C to −15 °C, and there is a large space for reducing the freezing cycle time. (2) In terms of buffer distance, Li et al. (2013) [13] used theoretical calculation and numerical simulation methods to calculate the buffer distance between the high-temperature mining area and the frozen wall curtain in detail, and determined a reasonable buffer distance, which greatly shortened the time of curtain closure and reduced the cost. (3) In terms of cold source, Zhang et al. (2012) [14] studied the feasibility of using a natural cold source in severe cold regions for frozen wall refrigeration, and found that although the freezing time is longer, the freezing cost can be greatly reduced.

Although the above studies analyzed the factors affecting the closure of frozen wall curtains and the cost of use from various perspectives, there is still a lack of clear understanding of the establishment of frozen wall curtains in water-bearing geological environments. In practical engineering, when the soil freezes under geological conditions under different water saturation conditions, the time between the formed freezing circle and the frozen wall curtain is very different, and even the frozen wall curtain cannot be closed under some conditions. Some scholars have found that the main reason for this is that the thermal conductivity of the water-bearing soil changes. Wang et al. (2001) [15] used an indoor simulation of a frozen soil test field and found that the water saturation of the overlying soil is the main factor leading to the change in the formation of thermal conductivity. The change in thermal conductivity will have a greater impact on the closure of the curtain. However, in previous studies, the effect of the water saturation of the overlying soil on the formation time of the freezing circle has not been reported.

In this paper, the influence of the overlying soil under different water saturation conditions on the formation of the frozen wall curtain is studied. The simulated experimental platform is used to perform similar simulation experiments and research the influence of overlying strata under different water saturation conditions on the formation of the frozen wall curtain in the in situ mining of underground oil shale. Based on finite-element simulation, the change in temperature during the formation of the frozen wall under different water saturation conditions is analyzed, and the accuracy of the experiment is further verified with the experimental results. This paper provides a theoretical basis for setting the freezing time and freezing range of the frozen wall curtain in different water-bearing geological environments. It is expected to reduce the cost of freezing, reduce the impact of oil and gas leakage on the environment, and promote the efficient development of oil shale.

## 2. Theory, Materials, and Methods

### 2.1. Theoretical Basis

All theories include: fluid mechanics basis, heat conduction basis, latent heat treatment, and convection process.

Fluid mechanics basis: The basic equations of fluid mechanics include mass conservation equation, momentum conservation equation, and energy conservation equation [16]:

The mass conservation equation is:

$$\frac{\partial \rho}{\partial t} + \nabla(\rho v) = 0 \tag{1}$$

where $\rho$ is the density, g/cm$^3$; $v$ represents the vector representation of the flow velocity at the center of the control volume;

The momentum conservation equation is:

$$\rho\frac{dv}{dt} = \rho F + \mu\nabla^2 v - \nabla p \tag{2}$$

where $F$ is the volume force in unit volume, KN; $\nabla p$ is the pressure difference between two seepage sections; $\mu$ is the viscosity of the liquid.

The energy conservation equation is:

$$\rho C\frac{dT}{dt} = k\nabla^2 T + Q \tag{3}$$

where $C$ is the specific heat capacity of rock mass, J/(kg·°C); $T$ is the temperature, °C; $Q$ is the total heat, J; $k$ is the permeability.

Heat conduction basis: The heat conduction theory concerns a large number of molecular thermal motions on the micro level of matter. The collision between molecules leads to the heat transfer of temperature from the part with high temperature to the part with low temperature. Therefore, the generation of heat conduction must have a temperature difference, and the rate of heat conduction also depends on the size of the temperature difference [17]. According to the Fourier law, in the process of heat conduction phenomenon, the heat passing through a fixed section in a unit of time and the temperature change rate perpendicular to the section are proportional to the section area, and the direction of heat transfer is the same as the direction in which the temperature decreases. Combined with three basic equations of fluid mechanics, the differential equation of heat conduction of the underground frozen wall can be obtained [18,19]:

$$\rho C\frac{\partial T}{\partial t} = \nabla(\lambda\nabla T) + S(x,y,z,t) \tag{4}$$

where $S\ (x,y,z,t)$ is the cross-sectional area, m$^2$.

Latent heat treatment: In the process of underground frozen wall curtain formation, due to the different water saturation conditions of rock mass, the water heat released in the rock mass when the liquid becomes solid is also different, which affects the time and speed used for the closure of the freezing rubber ring.

The existing methods for latent heat treatment in China and abroad include the enthalpy method, the temperature rebound method, and the equivalent specific heat capacity method [20]. Among them, the calculation process of the enthalpy method and the temperature rebound method on specific heat capacity is more complex. Therefore, the equivalent specific heat capacity method is used to deal with the latent heat phenomenon.

The equivalent specific heat capacity is to avoid the occurrence of latent heat in the calculation process by modifying the specific heat capacity. The modified specific heat capacity is:

$$C_X = \begin{cases} C_2, & T < 0\,°C \\ C_P, & 0\,°C \le T \le 2\,°C \\ C_1, & T > 2\,°C \end{cases} \tag{5}$$

where $C_1$ is the specific heat capacity of water, J/(kg·°C); $C_2$ is the specific heat capacity of ice J/(kg·°C).

$$C_P = C_2 + \frac{L}{I_1 - I_2} \tag{6}$$

where $I_1$ is the liquid temperature, °C; $I_2$ is the solid temperature. °C; $L$ is the specific heat capacity increase caused by latent heat, J/(kg·°C);

Convection process: Due to the different water saturation conditions of underground rock mass, fluid (water) also generates heat conduction. This phenomenon is called convection; that is, when the temperature difference is generated inside the rock mass, the

temperature difference makes the relative motion of each part of the liquid water, thereby realizing the heat transfer. Further, heat transfer is generated when the fluid flows in contact with the solid wall with a temperature difference, which is referred to as a convection heat transfer. It is a comprehensive result of heat conduction and heat convection. Therefore, during the underground frozen wall curtain formation, only the heat conduction equation inside the solid is considered. Conduction near the solid wall with temperature difference satisfies [21,22].

$$\lambda \frac{\partial T}{\partial n} = \alpha(I_1 - I_3) \tag{7}$$

where $I_3$ is the temperature near the solid wall, °C; $n$ is the normal vector near the solid wall; $\alpha$ is the convective heat transfer coefficient that is constant.

### 2.2. Temperature Experiment of Underground Frozen Wall

Since the main components of the stratum are sedimentary rocks and magmatic rocks, this experiment uses a mixture of coarse sand and fine sand, which is a similar composition to simulate the formation. The lower layer is an energy substance such as oil shale and marlite. The experiments only consider the feasibility of the in situ mining of underground oil shale, without considering cost issues and the subsequent heating and mining of oil shale. Therefore, the freezing range is only the upper simulated formation. The effects of the different water saturation conditions of the overlying strata on the curtain formation of the frozen wall were explored. The overlying strata with a water saturation of 10%, 20%, and 100% were selected for the simulation experiment and compared with each other.

Aiming at the influence of the different water saturation conditions on the formation of the frozen wall, design experiments were conducted to simulate the experimental monitoring of the surrounding temperature fields in the formation of the frozen wall. According to the demonstration experiment of the United States Shell ICP technology, the spacing between the two frozen walls was 2.4 m, and the outer diameter of the frozen wall was about 300 mm. In the experiment, the diameter of two frozen walls was designed to be 10 mm, with labels $L_1$ and $L_2$. Similar ratios of 1/30, according to the similarity criterion, of the two frozen walls were placed according to the distribution of the 80 mm center distance. According to the treatment of convection and the theory of the far radius, the external freezing soil specification was a 600 mm × 500 mm × 400 mm long square experiment tank filled with 50 mm of clay under the tank to simulate the lower oil shale. The upper layer was filled with a coarse and fine sand mixture to simulate the overlying strata. The temperature of the freezing soil box was monitored by 7 electronic thermometers and labeled D1–D7. Schematic diagrams of the temperature measuring hole and condenser tube arrangement are shown in Figure 1.

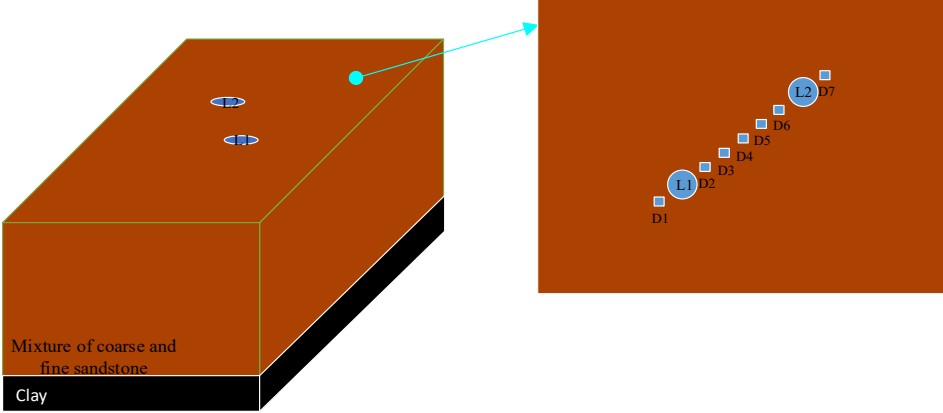

**Figure 1.** Experimental equipment layout schematic diagram.

There is only one variable in the experiment, namely, the water saturation of the overlying strata. Before the experiment, the sandy soil was placed in different water-

containing environments, and the water content of the sandy soil was repeatedly measured by the drying method in a constant temperature environment of 105°. The average value was taken to determine the water saturation of 10% and 20%. The water supply system was used to make the sandy soil surface have a thin water film, which meets the requirement of 100% saturation. It was covered with a preservative film to keep the water saturation constant. The freezing machine was set at −15 °C, and $L_1$ and $L_2$ were refrigerated at room temperature (20 °C) to determine the temperature changes at different positions. Studies have shown that the freezing effect is best when the center of the connection line of the frozen wall is at −4 °C [23]; therefore, the time when the D4 thermometer reaches −4 °C was used to characterize the time when the frozen wall of the two condenser tubes completed the circle, indicating the formation of the frozen wall curtain.

*2.3. Temperature Numerical Simulation of Underground Frozen Wall*

Since the experimental phenomenon of frozen wall curtain formation is relatively complicated, it is necessary to facilitate the establishment of the model. The following assumptions are made in the simulation:

(1) Heat transfer is only in the radial direction during the freezing process, and other directions can be ignored.

(2) It does not consider the impact of the ambient temperature; that is, all simulations are a constant temperature.

(3) The initial temperature in the formation is constant.

(4) The effects of depth on temperature can be ignored [24,25].

According to the above assumptions, the model can be simplified to a two-dimensional form, and the thermal conductivity $\lambda$ and the internal heat source $\dot{q}$ are set to constants. Equation (4) obtained from the above three basic equations of Fourier's law and fluid mechanics is used as the thermal conductivity differential equation of overlying strata.

Differential equations of the heat conduction of two frozen walls can be written as:

$$\frac{\mathrm{d}^2 t}{\mathrm{d}r^2} + \frac{n}{r}\frac{\mathrm{d}t}{\mathrm{d}r} + \frac{\dot{q}}{\lambda} = 0 \tag{8}$$

where $r$ is the length coordinate starting from the center; $\dot{q}$ is the internal heat source. W/m³; $n$ = 1. This can be solved by the following boundary conditions: when $r = 0, \frac{\mathrm{d}t}{\mathrm{d}r} = 0$; when $r = R$, it is the third boundary condition [26], that is, when the heat transfer medium contains water:

$$-\lambda \left( \frac{\mathrm{d}t}{\mathrm{d}r} \right)_R = \alpha (t_w - t_\infty) \tag{9}$$

where $R$ is the radius; $t_w$ is the temperature of the target position, °C; $t_\infty$ is the temperature of the fluid, °C;

The initial temperature of the soil was set to be 20 °C, namely

$$T|_{t=0} = 20 \tag{10}$$

According to assumptions, the distance was solved in an infinite place to be the same as the initial temperature, namely:

$$T|_{t=\infty} = T_\infty \tag{11}$$

ABAQUS is a powerful set of finite-element software for engineering simulations that solves problems ranging from relatively simple linear analysis to many complex nonlinear problems. ABAQUS includes a rich library of elements that can model arbitrary geometric shapes, and it has various types of material model libraries that can simulate the performance of typical engineering materials and can also simulate many problems in other engineering fields, such as heat conduction. It is very suitable for the temperature field of the underground frozen wall, so ABAQUS software is used for heat conduction

simulation [27]. In the simulation, only one degree of freedom was set, namely, water saturation. In order to make the simulation results more accurate, the upper and lower radius of the joint structure region was set as the radius of the far boundary. According to the far radius theory [28]:

$$R_\infty \geq 4\sqrt{a_s t} \tag{12}$$

where $R_\infty$ represents the infinite boundary radius; $a_s$ is the maximum radius of the temperature region time $t$.

The radius of the boundary is the radius of action at time $t$. Therefore, in the simulation, if the temperature of the far boundary radius is changed, that is, temperature stress is generated, then the far radius will continue to increase. Combined with the far radius theory and convection treatment, the free meshing method is adopted to ensure fine meshing under the condition of calculation accuracy. The grid was set to 0.03 mm in the solution area and dispersed into 40,120 units, as shown in Figure 2.

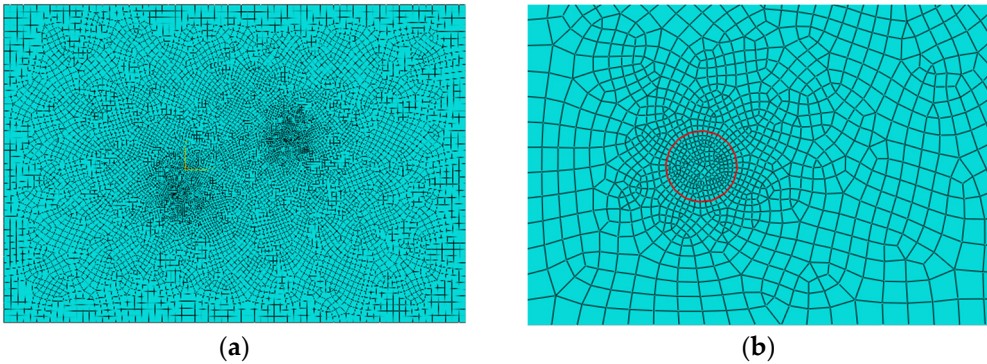

(a)                     (b)

**Figure 2.** Grid division. Red circle in (**b**) represents a frozen well. (**a**) Overall division; (**b**) Division around frozen well.

Since the experimental material was a mixture of coarse sand and fine sand, the thermophysical parameters set by the simulation were consistent with the actual experiment. The density of the water is 0.99 g/cm$^3$, specific heat capacity is $4.18 \times 10^3$ J/(kg·°C), thermal conductivity is 0.6 W/m·k, and viscosity is 0.001 kg/m·s. The melting heat of ice is $3.33 \times 10^5$ J/kg, and water condenses into ice to release the same heat.

Different water saturation conditions use thermal conductivity, specific heat capacity, and density to endue the model. The thermally conductive coefficient increases with an increase in water saturation and often uses the average thermal conductivity of the soil in practical engineering, as shown in Equation (13):

$$q = \frac{Q}{A} = \lambda \frac{dT}{dx} \tag{13}$$

where $A$ is the area of heat passing through, m$^2$.

The thermal conductivity of coarse and fine sandstone under different water saturation conditions is shown in Table 1.

**Table 1.** Actual thermal conductivity of sandstone under different water saturation conditions.

| Soil | | Water Saturation (%) | Thermal Conductivity (W/m·k) |
|---|---|---|---|
| Grit sandstone (1–2 mm) | Compact | 10 | 1.865 |
| | | 20 | 3.231 |
| | Loose | 10 | 1.165 |
| | | 20 | 2.741 |
| Fine sandstone (0.25–1 mm) | Compact | 10 | 2.558 |
| | | 20 | 3.935 |
| | Loose | 10 | 1.561 |
| | | 20 | 3.694 |

Therefore, when the water saturation is 10% and 20%, the corresponding average thermal conductivity is selected to replace. During the experiment, the simulated oil shale covered with 100% water saturation was in the way of water injection, there is no actual measurement data of thermal conductivity in practical engineering; therefore, the thermal conductivity of 100% water saturation can be calculated directly by Equation (13). In the simulation, the gas composition in the experimental materials was ignored for the difference in water saturation in this experiment. The specific heat capacity $C_P$ of simulated sand at different water saturation conditions between 0 °C and 2 °C is calculated according to the above equivalent specific heat capacity (5) and Equation (6), the specific heat capacity of rock mass in other states were replaced by $C_1$ and $C_2$. The density of sand under different water saturation conditions is generally between 1.4 g/cm$^3$ and 2.3 g/cm$^3$, the density of saturated sand was between 1.8 g/cm$^3$ and 2.3 g/cm$^3$. According to the theoretical value and the experimental material parameters, the reasonable parameters of different water saturation conditions were set as shown in Table 2 [29].

**Table 2.** Different water saturation model parameters.

| Water Saturation (%) | 10% | 20% | 100% |
|---|---|---|---|
| Thermal conductivity (W/m·k) | 1.795 | 3.400 | 14.370 |
| Heat capacity (J/(kg·°C)) | $1.24 \times 10^3$ | $1.47 \times 10^3$ | $2.91 \times 10^3$ |
| Density (g/cm$^3$) | 1.52 | 1.67 | 2.03 |

Since this experiment does not consider the problem of groundwater flow rate, the set result for the convective heat transfer coefficient was low, and the value was 200 W/m·k. In the experiment, when the D4 thermometer reaches −4 °C, the soil is considered to be freezing; therefore, the simulation was set to −4 °C as the freezing temperature. That is, when the temperature at the midpoint of the connection between the two freezing holes reaches −4 °C, it is considered that this freezing has the ability to make the soil around the freezing hole form a freezing curtain. The initial calculation can be stopped, as shown in Figure 3.

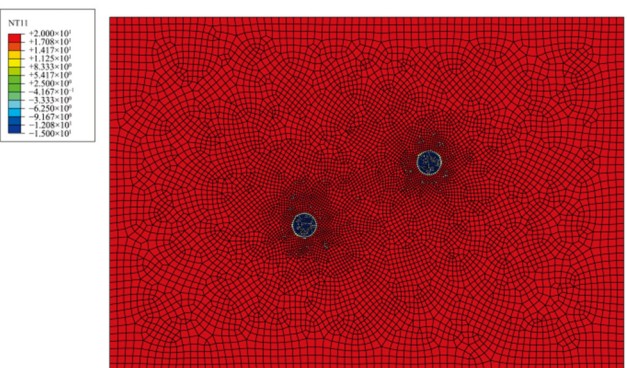

**Figure 3.** Simulation initial calculation diagram.

## 3. Results and Analysis of Experiments and Numerical Simulations

### 3.1. Experimental Results and Analysis

The temperature changes of the D1–D7 electronic thermometers were recorded at different times in the experiment, and curves were drawn, as shown in Figure 4.

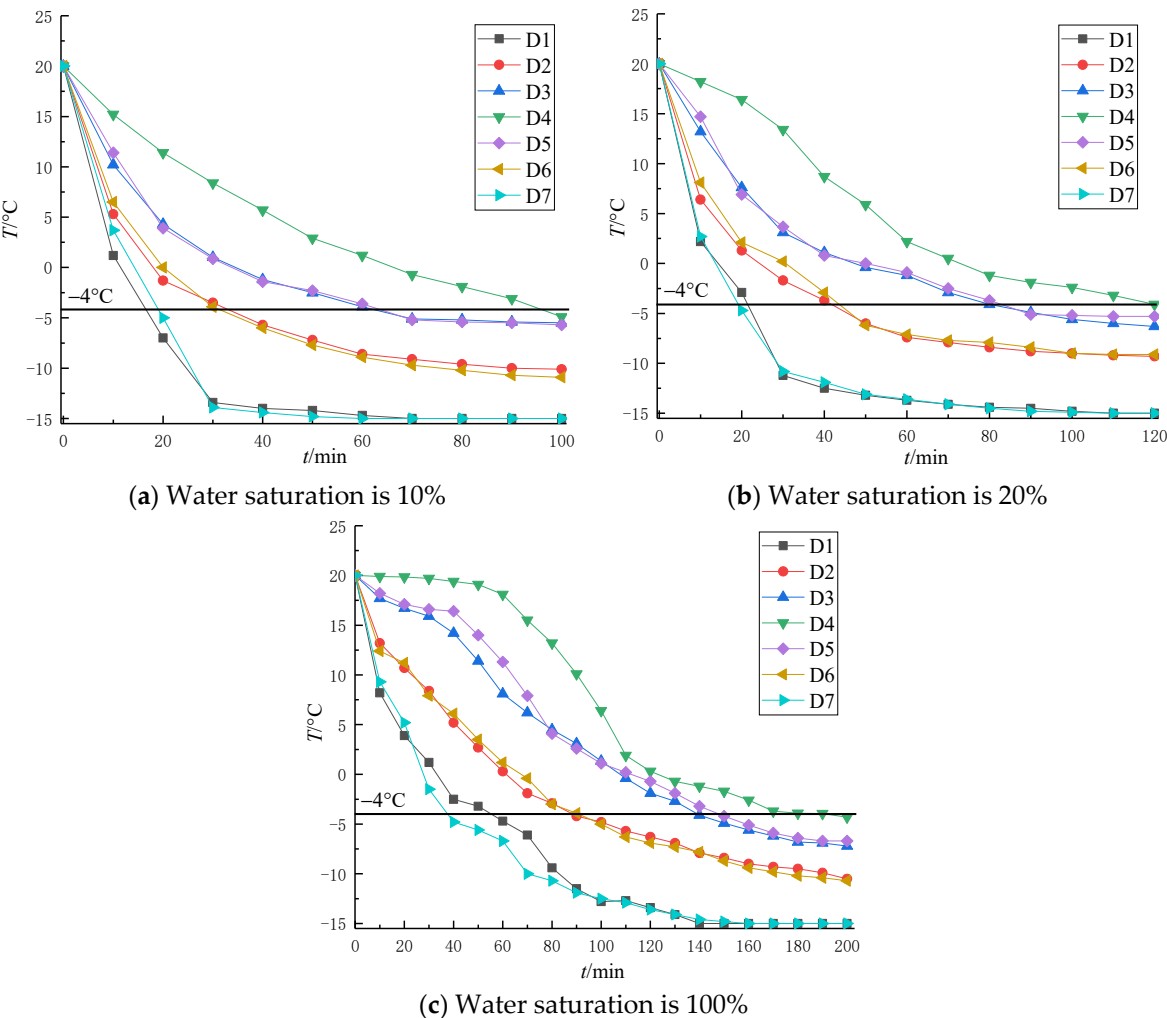

(**a**) Water saturation is 10%　　　　　(**b**) Water saturation is 20%

(**c**) Water saturation is 100%

**Figure 4.** Temperature changes of electronic thermometers under different water saturation conditions.

According to Figure 4, under the same water saturation condition, with the increase in time, the closer to the two frozen walls, the faster the temperature drop rate, and the temperature drop rate was the slowest at the central position (D4) of the connection between the two frozen walls. At the same freezing time, with the increase in water saturation, the temperature decrease rate at all positions decreased, the initial decrease rate became more gentle, the time required for the temperature of D4 to decrease to −4 °C was gradually extended, and the time taken was calculated, as shown in Table 3.

**Table 3.** Time taken for the temperature of D4 to drop to −4 °C under different water saturation conditions.

| Water Saturation (%) | 10% | 20% | 100% |
|---|---|---|---|
| Freezing time (min) | 95.2 | 118.7 | 192.4 |

A fitting curve of the time taken for the D4 temperature to drop to −4 °C under different water saturation conditions was drawn, as shown in Figure 5.

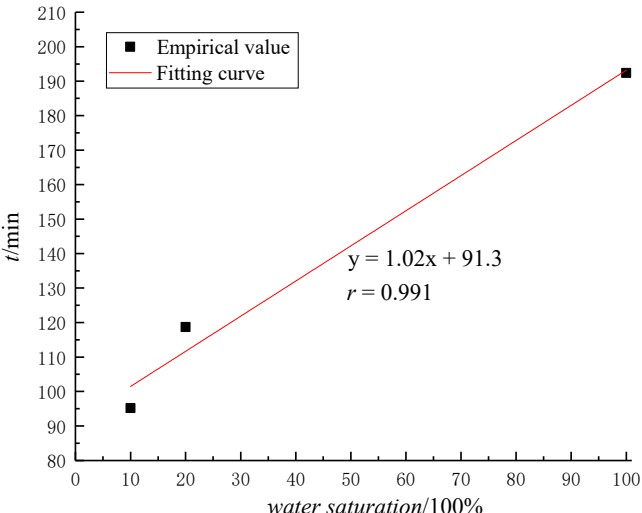

**Figure 5.** Relationship between different water saturation conditions and time taken for D4 temperature to drop to −4 °C.

According to Table 3 and Figure 5, it can be seen that under different water saturation conditions, the freezing time at the center of the connection line of the frozen wall was prolonged. In addition, sandy soil is a porous medium, and water, air, and heat in sandy soil are interrelated. Ignoring the effect of gas, purely from the point of view of thermodynamics, the transfer of heat in sandy soil causes water migration, and water migration is accompanied by heat transfer. The initial temperature field of the soil during the freezing process changes and causes water phase transition; that is, liquid water gradually transforms into solid ice, and the freezing radius (phase interface) of the heat conduction process will expand over time. With the increase in water saturation, the temperature decrease in the initial time of all positions gradually slowed down; that is, the water saturation restricts the formation of the frozen wall closure, which leads to the extension of the freezing time, and the longest time is when the water saturation reaches 100%.

### 3.2. Numerical Simulation Results and Analysis

The model was calculated and solved. The time required for the soil temperature to drop to −4 °C at the midpoint of the line connecting the two freezing holes with a water saturation of 10% was 62.7 min. The temperature diagram of other models at 62.7 min is shown in Figure 6.

According to the simulation results, after 67.2 min of low-temperature heat transfer in the model with a water saturation of 10%, the center of the connection line between the two frozen wells reached −4 °C, and it can be considered that the frozen wall between the two frozen wells was formed. Under the same time conditions, the temperature at the center of the line connecting the two frozen wells in the model with a water saturation of 20% was about −1 °C; the temperature at the center of the line connecting the two frozen wells in the model with a water saturation of 100% was about 14 °C. This shows that when the heat transfer time is the same, the greater the water saturation, the slower the heat transfer rate, and the smaller the influence range of the two frozen wells. The influence range of the model with a water saturation of 100% after 67.2 min of low-temperature heat transfer is only one-third of that of the model with a water saturation of 10%. Therefore, the water saturation restricts the heat transfer rate and the influence range. The higher the water saturation, the more serious the restriction. We continued calculating 50% and 100% and obtained the temperature change curve of the center position of the two freezing holes under different water saturation conditions, as shown in Figure 7.

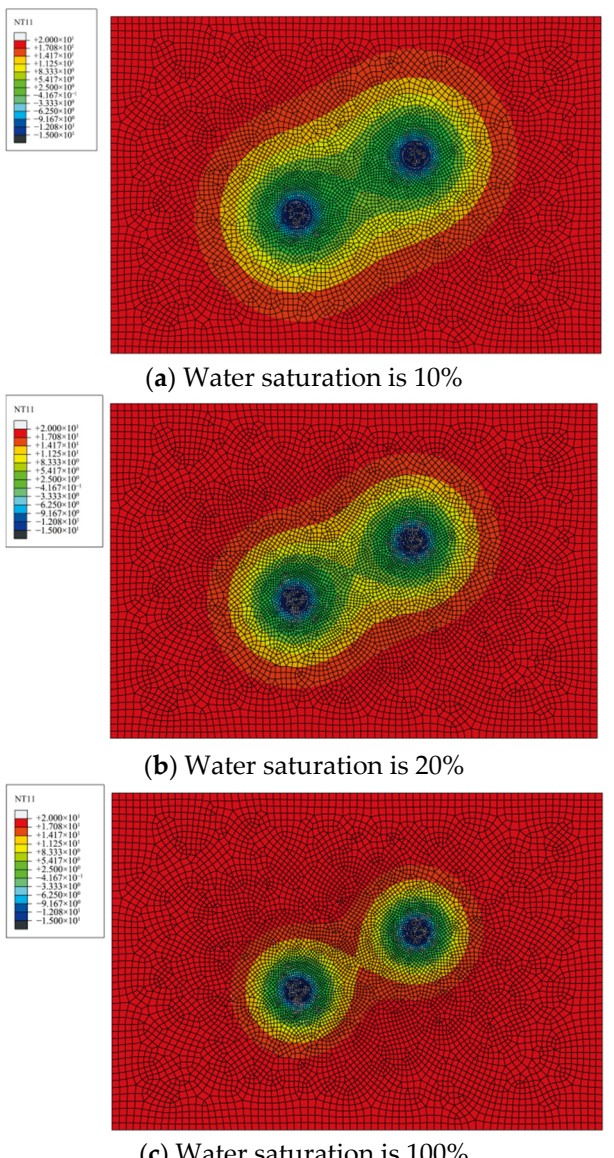

(**a**) Water saturation is 10%

(**b**) Water saturation is 20%

(**c**) Water saturation is 100%

**Figure 6.** Schematic diagram of heat transfer temperature distribution after 62.7 min under different water saturation conditions.

According to Figures 6 and 7, when the heat transfer is 62.7 min in different water saturation models, the water saturation of the 50% and 100% models did not decrease to $-4\,°C$ at the center of the connection between the two freezing holes. Among them, the temperature at the central position with a water saturation of 50% is 6.41 °C, the temperature of the model with a water saturation of 100% is 11.8 °C, and the temperature decline trend of the frozen wall connection center is different under different water saturation conditions. Under unsaturated conditions, the initial temperature decreases rapidly, and with the increase in water saturation, the initial cooling gradually becomes gentle. When the water saturation is 100%, there is even no temperature change within 10 min. This shows that the water saturation restricts the formation state and time of the frozen wall. The higher the water saturation, the longer the freezing hole temperature decreases. The time length used when the temperature at the central position of the simulated frozen wall connection is reduced to $-4\,°C$ under different water saturation conditions is shown in Table 4.

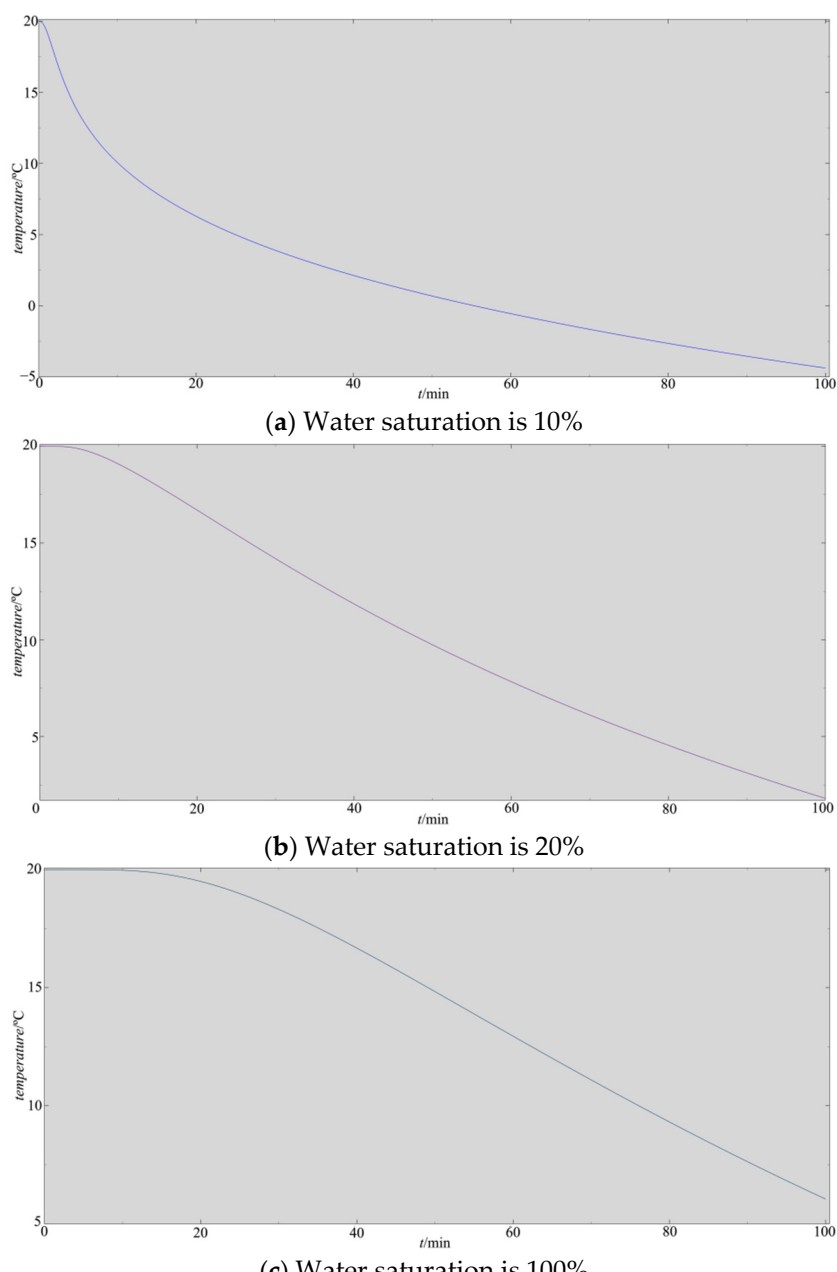

(**a**) Water saturation is 10%

(**b**) Water saturation is 20%

(**c**) Water saturation is 100%

**Figure 7.** Temperature variation curve of the center position of the connecting line between two frozen holes under different water saturation.

**Table 4.** Freezing time at the center of the connection line of simulated frozen wall under different water saturation conditions.

| Water saturation (%) | 10% | 20% | 100% |
|---|---|---|---|
| Freezing time (min) | 62.7 | 103.4 | 177.7 |

The freezing time and fitting optimal curve under different water saturation conditions are shown in Figure 8.

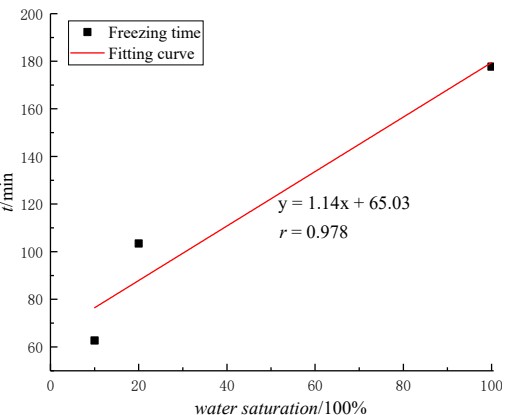

**Figure 8.** Relationship between different water saturation and freezing time.

According to Table 4 and Figure 8, with the increase in water saturation, the time to reach $-4\ °C$ at the center of the connection between the two frozen walls gradually increases. Compared with the experimental data and the slope of the curve, there is a certain error in the freezing time obtained from the simulation and the slope of the fitting curve, and the simulation results are all lower. This is because to facilitate the establishment of the model, the model is set to a two-dimensional plane model. The heat transfer in other directions, the influence of the external environment on the temperature of each position, and the influence of depth on temperature and freezing effect are ignored. Moreover, the temperature drop of the established model is relatively uniform, and the obtained curve is also relatively smooth. The data measured in the experiment have heat transfer in other directions and have the influence of the external environment and depth; therefore, there are some differences in the distribution of temperature in the experimental results, and the curve also fluctuates. However, the variation shows little difference in the change trend, indicating that the model was established under the assumed conditions, which was consistent with the experimental results.

In addition, the temperature difference $t_0 - t_w$ can be calculated according to the above-mentioned boundary condition that the heat transfer medium is rich in moisture:

$$t_0 - t_w = \frac{\dot{q}R^2}{m\lambda t_\infty} \qquad (14)$$

where, $t_0$ is the temperature of the heat source, $°C$; $m = 2n + 2$ is a constant.

According to Equation (14), the fluid temperature $t_\infty$ is inversely proportional to the temperature difference, The lower the temperature of $t_\infty$, the greater the temperature difference. With the increase in water saturation, the longer it takes for the temperature of water to decrease from normal temperature ($20\ °C$) to $-4\ °C$, the slower the rate of decrease, resulting in a smaller temperature difference; that is, the temperature $t_w$ at the center of the line connecting the two frozen walls decreases slowly [30].

Considering the change in temperature in the actual experiment, the thermal conductivity $\lambda_h$ of water will also change, so it is still necessary to consider the influence of the change in thermal conductivity $\lambda_h$, The thermal conductivity $\lambda_h$ increases with the increase in temperature. Under atmospheric pressure, the thermal conductivity of water can be calculated by Equation (15)

$$\lambda_h = 0.58(1 + 0.002t_h) \qquad (15)$$

where $t_h$ is ambient temperature, $°C$.

It can be seen from Equation (15) that with the low-temperature diffusion of the frozen wall, the thermal conductivity of water also decreases; that is, the thermal conductivity of water is positively correlated with the decrease in temperature. This is due to the lower specific heat capacity of water at low temperatures. Thus, the diffusion rate of temperature

in water-rich soil slows down, resulting in a longer freezing time at the center of the frozen wall connection.

The water saturation has a great influence on the formation of the freezing circle and the frozen wall curtain. The freezing time becomes longer as the water saturation is higher. When the water saturation is 100%, the freezing rate is the slowest. These results will provide a safe theoretical and experimental reference for the establishment of the frozen wall curtain in the in situ mining of underground oil shale.

## 4. Conclusions

The frozen wall curtain is an important step in the in situ mining of underground oil shale. It plays an important role in sealing water, oil and gas leakage prevention, and even support. However, in practical engineering, the curtain often has difficulty in closing, and the freezing cost remains high. Based on this, we studied the influence of overlying soil under different water saturation conditions on curtain closure, which is an important reason for the difficulty of curtain closure. By means of a combination of experiments and numerical simulations, the closure law, time used, and influence range of the overlying soil frozen wall curtain under different water saturation conditions were studied. Aiming to reduce the freezing cost of frozen walls, the main results are as follows:

(1) With the increase in water saturation, the longer the freezing time at the center of the connection line of the frozen wall, the more difficult it is to form a freezing circle, and the worse the freezing effect.

(2) The temperature change trend of the center point is different under different water saturation conditions. The higher the water saturation is, the slower the initial temperature drop rate is. When the water saturation is 100%, the initial temperature drop hysteresis phenomenon occurs. In the experiment, when the water saturation is 100%, the freezing circle is gradually formed until 194.2 min.

(3) The freezing time of the temperature conduction process model established in the simulation experiment is in good agreement with the experimental results. As the water saturation increases, the influence range of the two frozen walls at the same time gradually decreases. Therefore, water saturation is an important factor for the formation of the frozen wall curtain, which is of great significance to the in situ mining of oil shale in practical engineering.

**Author Contributions:** Conceptualization, H.L. and L.W.; methodology, L.W.; software, J.L.; validation, H.L., L.W. and J.L.; formal analysis, T.R.; investigation, T.R.; resources, T.R.; data curation, T.R.; writing—original draft preparation, J.L.; writing—review and editing, J.L.; visualization, J.L.; supervision, H.L.; project administration, L.W.; funding acquisition, L.W. All authors have read and agreed to the published version of the manuscript.

**Funding:** This research was funded by the National Key R&D Projects (Grant No. 2017YFC1503101), the General Programs of National Natural Science Foundation of China (Grant No. 51704142), and the Liaoning Province Doctoral Fund Project (Grant No. 2019-BS-115).

**Institutional Review Board Statement:** Not applicable.

**Data Availability Statement:** The data used to support the findings of this study are included within the article.

**Acknowledgments:** All individuals have consented. We thank Ziheng Zhang for their support with part of the implementation of experiments. We thank Siyang Sun for their support with part of the language correction. All authors have given their consent to the publication of the manuscript.

**Conflicts of Interest:** The authors declare that there is no conflict of interest.

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
