# Peer review of "Influence of Water Saturation on Curtain Formation of Frozen Wall"

_processes, doi:10.3390/pr10102065_

Round 1

Reviewer 1 Report

The manuscript is about the influence of water saturation on curtain formation of frozen wall. The scope of this article is consistent with the requirements of the Processes Journal, but it requires major revision in accordance with the comments below:

1.     The abstract is too long. According to the Instructions for Authors the abstract should be a total of about 200 words maximum.

2.     Avoid linking references in Introduction and Theoretical basis as in: [4-6], [8-10], [16-18].
Please combine no more than two references. Instead summarise the main contribution of each referenced paper in a separate sentence.

3.     Research manuscript should comprise sections such as: Introduction, Materials and Methods, Results, Discussion, Conclusions. Please rearrange the article according to the guidelines (Instructions for Authors). Materials and Methods should be described with sufficient detail to allow others to replicate and build on published results. Sections Results and Discussion could be combine and in this section should be description of experimental results and discussion of them. Therefore points: 2.1-2.4, 3.1, 4.1-4.2 should be in Section Materials and Methods and points 3.2 and 4.3. in Section Results and Discussion.

4.     Poor quality of figure 2, is too small.

Reviewer 2 Report

Dear Authors,

I have read your work with great interest. The work deals with an issue with great application potential in the context of considerations based on computer calculations. I would like to share my comments on the performed calculations. I encourage you to consider the questions posed and, if possible, to include them in the revised version of the work.

1. Abstract - factually correct. It does not answer the question, what is the mathematical form of the model under consideration (ODEs, PDEs, DAEs)?

2. Keywords - I encourage you to add "numerical simulations", or something that indicates that the work also touches on the issue of computer simulations.

3. Introduction line 40: The citation [4-6] is too general.

4. Theoretical basis - equations (1) - (3) are left without comment. The equations and variables should be explained right away, not until the end of section 2.2

5. Section 2.3: Important question - is Cx (T) (eq. 5) continuous with temperature? Is it differentiable? How does this affect the computation?

6. Figure 1 - was it made using specialized software?

7. Figure 3 - why 3 points were taken into account. What is the standard deviation of the resulting linear regression equation?

8. Section 4.1 line 241: ABAQUS software - why is this software, no reference to the description of the calculation procedure operation.

9. Section 4.3: careful analysis of the obtained results, very good graphic presentation (Figure 6). 

10. Conclusion - what are the perspectives of your future research work?

11. References - correct selection of literature sources.

I believe that the work can be published after considering the indicated  issues and questions.

Reviewer 3 Report

Oil shale is a mineral that has recently become an important raw material for the production of shale oil.

There are two ways of extracting shale oil from oil shale, associated with the use of ground and underground distillation.

The traditional, mine method of developing shale deposits has been known for a long time and is not a big problem. However, its use is appropriate only in cases where deposits are located near the surface of the earth.

The disadvantages of this method are a significant impact on the relief of the surrounding area and the release of a huge amount of carbon dioxide during the further distillation of raw materials by pyrolysis. Although now there are new methods to capture CO2 in significant quantities, but in general the problem remains unresolved.

The second method is much more expensive and more complicated, since it involves drilling numerous wells, hydraulic fracturing, thermal and chemical effects. In addition, it requires the involvement of a huge number of specialists and equipment, as well as a lot of research.

And at the same time, it does not guarantee long-term highly productive operation of the well. As a rule, after a year of operation, there is a significant decrease in oil production. You have to either repeat the whole procedure from the beginning, or stop operation.

Naturally, such a devastating impact on nature causes opposition from environmental organizations and leads to a legislative ban on such mining methods. However, despite all the difficulties that arise, this method is constantly being improved and is widely used in many countries.

The extraction of gas from oil shale by the method of inclined horizontal drilling is profitable only if there are large areas of occurrence of this mineral. The advantage of this method lies in conducting deep hydraulic fractures, creating cracks and further pumping out gas, which is most convenient when fields are located near densely populated areas that do not have other energy resources.

However, a large set of environmental problems and the high cost of this method create significant difficulties in its implementation.

The use of oil shale as a fuel is inefficient: little heat is released, but soot is an excessive amount.

It is much more reasonable to get fuel from them in the form of oils and resins. In addition, gasoline, household gas, and a number of substances in the chemical industry are obtained from this class of minerals. Aromatic hydrocarbons, benzene, coke, adhesives, plastics, synthetic tanning agents, herbicides, bitumen, and even medicines contain substances derived from oil shale.

China is among the leaders in shale oil and shale gas reserves.

 Therefore, the topic of research is interesting and relevant.

Remarks

In my opinion, the Introduction is not written clearly, chaotically.

The article does not highlight the scientific novelty of research, practical value.

It is not clear from the conclusions and it is not clear how the results of the studies will be applied in mining practice.

Not all formulas contain decoding symbols (1, 2, 3, etc.)

In expression 5, the signs , >,

Line 103 and 136, 229 repeat the same designation

Line 135 T - temperature of what? I1 is?

Line 245 where as ?

Round 2

Reviewer 1 Report

It can be accepted in the present form.